# The impact of reusable tableware packaging combined with environmental propaganda on consumer behaviour in online retail

**Chao Gu[1], Jiangjie Chen[2], Wei Wei[3], Jie Sun[1], Chun Yang[4], Liao Jiang[5], Jingyue Hu[5], Baiwan Lv[5], Shuyuan Lin[6], Qianling Jiang[4]***

1 Department of Culture and Arts Management, Honam University, Gwangju, Korea, 2 The Graduate Institute of Design Science, Tatung University, Taipei, Taiwan, 3 School of Textile Garment and Design, Changshu Institute of Technology, Changshu, China, 4 School of Design, Jiangnan University, Wuxi, China, 5 School of Art and Design, Minnan Science and Technology University, Quanzhou, China, 6 Department of Media Design, Tatung University, Taipei, Taiwan

\* jiangqianling@jiangnan.edu.cn

**Data Availability Statement:** All data files are available from the Dryad database URL: https://datadryad.org/stash/share/HQiN1DTk9HLLQE_

## Abstract

With the development of the takeaway industry, the demand for disposable cutlery is increasing, posing a heavy burden on the environment. Helping reusable tableware increase market share is important because it helps preserve the natural environment while making commercial gains. Given the additional cost to consumers of using reusable tableware in many settings, this article examines the impact of incorporating environmental propaganda into packaging design on consumer behaviour. The results show that the new packaging with high environmental propaganda satisfaction improves consumers' brand loyalty, purchase intention and continuance intention. The packaging of low environmental propaganda satisfaction may have negative effects and should be used with caution.

## Introduction

In recent years, with the rise of China's retail takeout industry, a large proportion of catering consumption relies on disposable cutlery. The act of buying a meal using disposable cutlery has become an accepted part of life for many people with a takeaway habit. Disposable tableware has taken a heavy toll on the environment as the catering economy has improved [1]. In many cases, there is a strong demand for disposable boxes for food transportation and cutlery transportation, even when customers tell the store, they have their own tableware. Switching to reusable tableware imposes additional communication and shipping costs on consumers. At the same time, the customer also needs to bear the loss and risk that this may bring. In this context, both from the perspective of business and environmental protection, the key is to increase the willingness of consumers to pay additional costs. The preparation and use of disposable tableware have obvious negative impacts on the environment, including multimaterial meal boxes and disposable chopsticks or spoons. In the case of fast-growing poplar chopsticks, China's annual consumption of 45 billion pairs is 1.66 million cubic metres of trees [2]. The proliferation of fast-growing forest plantations may increase the risk of drought in certain

t3JFdMmJrzvrh-DbmUNZyCEASvgI. DOI:https://
doi.org/10.5061/dryad.ns1rn8pvc This information
will only be available after acceptance.

**Funding:** The funders had no role in study design,
data collection and analysis, decision to publish, or
preparation of the manuscript.

**Competing interests:** The authors have declared
that no competing interests exist.

areas because groundwater pumping promotes transpiration. In Taiwan, the situation is even worse, with a population of 23 million consuming 17.5 million disposable tableware sets a day [3]. Residents' reliance on disposable cutlery is having an impact on global warming. Disposable used cutlery left in the soil can take years to decompose. Even if the cutlery is ideally delivered to an incineration site for incineration and subsequent landfill, it may cause air pollution and occupation of the landfill area. According to UKEA, cotton tote bags would need to be used at least 327 times to save energy and reduce emissions compared to high-density polyethylene (HDPE) bags. Reusable items may not always be the most logical answer for reducing carbon emissions. When the eco-friendly value of reusable items is based on reusing them enough times, it is worth discussing the impact of eco-promotional ideas on consumers' intentions to choose reusable tableware. There is a significant positive correlation between enterprises' social responsibility activities related to environmental protection and consumers' purchase intention (PI) [4]. This suggests that companies linking themselves to environmental themes will increase the chances of consumers buying goods. Consumers are more willing to compromise the quality and price of green products [5]. Under the same objective conditions, consumers will be less sensitive to the products' price and quality requirements with environmental protection attributes. Incorporating environmental protection propaganda into the design of a product may reduce the rigour consumers expect from the product. Creating a green corporate image may bring many advantages in business competition, which even gives rise to many greenwash behaviours that cannot protect the environment but hope to gain profits [6]. There has been little research on whether reusable tableware packaging combined with environmental promotion can help in retail. Reusable tableware is widely recognized as being more environmentally friendly than disposable cutlery, and using reusable tableware is environmentally beneficial in the subjective judgement of consumers. After being repeated enough times, reusable tableware is more cost-effective in terms of carbon emissions, so it is also environmentally friendly, objectively speaking. This paper takes ceramic chopsticks as an example to discuss the design effect of chopstick packaging combined with environmental protection elements. The influence of different packaging design effects on consumer brand loyalty (BL), PI and continuance intention (CI) was tested by adjusting the environmental propaganda content. It is urgent to examine the relationship between environmental advocacy and consumer behaviour in the packaging of ceramic chopsticks. Because it is good for business and good for the environment. On the one hand, through this study, we can give rationalization commercial advice on the packaging design of reusable retail tableware. On the other hand, more sales of reusable tableware will help protect the environment by reducing disposable cutlery.

## Literature review

### Ceramic chopsticks

In the widely used ceramic fabrication technique, the first step is to complete the 3D decoration and the clay preparation of the main ware, the design of the composition of the mould with the relief-based ware, and then the drying and biscuit firing. The second step is to add two-dimensional painting decoration to form a paint-based shape of the object, glaze it, and then enter the glaze firing process. The last optional step is the further decoration of on-glaze glaze, adding part of the ornament and on-glaze glaze firing to enhance the overall aesthetic level [7]. Because more than one calcination is required, the production of ceramic chopsticks consumes considerable energy. Carbonate in the kiln after 800-2000°C high-temperature reaction will release carbon dioxide [8]. This means that while ceramic chopsticks are green as reusable tableware, they need to be reused enough times to save energy and reduce emissions

in real life. Sun, et al. [9] measured and estimated tableware carbon emissions, and the results showed that reusable plastic tableware (RPT), cervical tableware and straw tableware had free carbon emissions of 7.00 g/Fu, 9.55 g/Fu and 14.6 g/Fu, respectively. Ceramic chopsticks, which produce a certain amount of emissions, are cutlery that falls somewhere between more environmentally friendly and higher emission. This is one of the reasons why this paper chooses it as a representative case for reusable tableware study. On the positive side, there are already people experimenting with energy-efficient fabrication methods in the ceramics industry. Gol, et al. [10] argue that using 3% weight glass waste instead of glass is an appropriate and sustainable practice. Chuenwong, et al. [11], proposed that the dry process of ceramic production could save 70% energy compared with the wet process through monitoring 28 kilns and 11 spray dryers. It is expected that with further technological innovation, carbon emissions from ceramic production will be reduced in the future. Finding ways to promote wider sales of ceramic chopsticks is good for business and the environment.

## Environmental Propaganda (EP)

Since 2011, China's Environmental Protection Bureaus has increasingly emphasized the importance of Environmental Protection and implemented several measures to control the environment. To raise public awareness of environmental protection, increasing publicity is an important part of the strategy [12, 13]. Chinese people have a high degree of acceptance of public interest publicity to promote the environment. Most people are familiar with the content and very supportive of the publicity [14]. This shows that the public has realized the importance of environmental protection in the long-term propaganda process. The behaviours that contribute to the environment are associated with positive perceptions. This kind of positive association is often applied in business behaviour. A study from the United Arab Emirates shows that green marketing has a positive and significant impact on consumers' environmental behaviour [15]. Research from Jordan shows that green marketing can significantly help develop nonprofit organizations [16]. Kumar, et al. [17] believe that in the clothing sales field, when the product or behaviour has the attribute of promoting environmental protection, consumers' recognition of environmental protection behaviour can be transformed into a favourable impression of the product, thus helping to improve the commercial effect. Compared with the products without the main environmental protection concept in the tableware market, the products that fully explain the environmental protection ability on the publicity may also gain stronger market competitiveness. It is worth noting that strained links may lead to consumer confusion and increased perceived risk to the product [18]. This kind of failed green marketing is regarded as the green wash of the brand by consumers, which will not only affect sales and profits but also cause moral and word-of-mouth damage [19]. Multiuse cutleries need to be used enough times to truly benefit the environment. This means there is a risk of being mistaken for green wash. In this context, it is useful to examine the impact of the use of the EP concept in marketing on reusable tableware brands.

## Packaging design

Packaging design is a powerful marketing tool in the retail environment, and consumers' decision-making process is strongly influenced by product packaging [20]. Packaging design can transform potential consumers into buyers, and consumers make 90% of purchase decisions after checking the packaging of products [21]. Packaging can shift consumers' attention, increase their understanding of value, improve their perception of product functionality, provide different shopping experiences and bring about different postshopping reactions [22]. Packaging design affects consumer perception and behaviour and is very important in

marketing. According to a survey, 60 percent of consumers tend to think packaging design is the most important factor in a new product, while only 17 percent think price [23]. Consumers think the package is even more important than the price. Previous research has shown that brands can bring multiple sensory experiences to consumers through packaging design [24]. For example, visual design elements in packaging should be attached to improve consumers' happiness [25] and influence consumers' taste perception through food packaging design [21]. Sustainable packaging is a relatively new concept that has received much attention in recent years [26]. In recent years, global consumers have become increasingly interested in EP, and green packaging is of great significance to the sustainable development of the environment [27]. For example, try to study and use eco-green packaging, eco-friendly packaging, sustainable packaging or recyclable packaging materials [28]. Several studies discuss the supply chain of green packaging [29], the new model of the circular economy brought about by green packaging [30], and the impact of changes in packaging materials on consumer behaviour [31]. To promote environmental improvement, brands can use several methods, including packaging design, to highlight the importance of EP to consumers. While most consumers are not inclined to read sustainability reports, they are receptive to the message of green responsibility in packaging [32]. This shows that packaging design is an effective means of conveying information. Consumers attach importance to environmental protection in packaging design and are willing to pay a higher price for low-carbon or environmental issues [33]. Whether this kind of consumer behaviour is the same for tableware brands needs further testing. Therefore, it is urgent and important to study the marketing effect of applying EP elements in multiplex tableware packaging.

## Environmental Propaganda Satisfaction (EPS)

An increasing number of consumers are beginning to prefer sustainable products [34]. Green concepts can help market some products. For example, adding EP in automobile publicity can effectively help to establish a good brand image (BI) and improve performance [35]. Environmental concern, green trust and products' functional values are the most influencing factors in purchasing energy-efficient home appliances [36]. However, many studies have shown that promoting green concepts in the marketing of some products is ineffective. For example, promoting environmentally friendly attributes does not help sales of cosmetics or cleaning products [37]. In the luxury industry, green marketing can enhance consumers' interest, but there is no direct correlation between it and PI [38]. Changes in consumers' EPS to brand marketing may be one of the reasons why green marketing sometimes helps and sometimes does not. A common example is when the concept is not expressed as expected, resulting in lower EPS. Furthermore, consumers may be less willing to buy products [39, 40]. Previous studies have shown that packaging affects consumer satisfaction with food [41]. Under business background, green packaging will affect consumers' satisfaction with products [42]. These studies show a close relationship between packaging and satisfaction. Satisfaction is considered the fulfillment the consumer receives when the service is provided [43]. When receiving a service, a consumer may compare the current service to a previous service [44]. If consumers' preset expectations are met, they may obtain a higher level of emotional response, and if they fail to meet expectations, they may respond with low satisfaction [45]. Therefore, for ceramic chopsticks, EP packaging and consumers on the brand environmental expectations of the comparison determine EPS. The effect of consumers' EPS with green packaging on brand marketing of ceramic chopsticks needs to be further verified.

## Brand image

BI is a multidimensional concept integrating consumers' quality, value, attitude, associations and feelings towards brands [46]. BI enables consumers to give an overall evaluation of various characteristics of a brand in a short time [47]. BI is considered consumers' overall perception and impression of the brand [48]. It summarizes the general perception of the consumer after contact with the brand. Brand image includes cognitive factors and affective factors [49]. Consumers' rational views and perceptual feelings towards the brand jointly reflect the overall BI. A positive BI needs a series of characteristics, such as innovative, focused, passionate, consistent, flexible, competitive, leadership and distinction [50]. When brand-related messages appear, consumers can connect products and brands through symbolic links [51]. Changing BI may directly affect consumer satisfaction [52]. At the same time, in many previous studies, it has been proven that there is a significant causal relationship between BI and PI [53, 54]. The influential effect of packaging on marketing would be affected by the brand. Therefore, it is necessary to discuss the BI separately.

## Brand loyalty

Inegbedion and Obadiaru [55] believed that BL is a behavioural reaction of consumers with preferences after thinking when facing multiple brands. Specifically, on the premise that there is more than one brand to choose from, consumers allocate different proportions of resources to these brands. Still, they have more resource preference or continuous purchase behaviour for specific brands [56]. Higher BL gives brands a greater chance of being selected on the shelf. BL is built upon long-term relationships between consumesr and brands [57]. BL is measured through psychological attachment and attitudinal advocacy [58]. The brand's core assets are closely related to external information, which represents consumers' attachment to the brand. Consumers with high BL are more willing to buy the same brand repeatedly in the retail market [59]. At the same time, BL represents a user's commitment to repurchase the product in the future [60]. BL is important in a retail business. High BL can make Brand appear in the minds of consumers as a priority choice when there is purchase demand [61]. Consumers' BL towards the brand can be comprehensively measured from behaviours such as repurchase behaviour, the share of wallet and quantity of brand purchases, and frequency of purchase [62]. Therefore, BL is used to test consumers' preference and purchase preference for ceramic chopstick brands.

## Purchase intention

PI is a consumer's intention to purchase a product or service [63]. Consumers' PI to retail goods is affected by many factors, such as marketing method, product type, and physical environment [64]. For example, green marketing is one of the important factors that may affect consumer PI [65]. Some brands try to associate products with positive green concepts. Consumers' positive feelings towards goods will influence their PI [66]. Previous studies have shown that consumers' PI can be increased by arousal, pleasure, positive emotion, and positive mood [67]. PI is used to predict consumers' future purchase likelihood [68]. The higher the PI is, the higher the possibility of consumers' purchasing behaviour. In addition to the possibility of purchase, PI also measures consumers' interest in products [69] and their specific purchase behaviour in the future [70]. PI is a process that starts from belief, goes through attitude and intention and finally forms behaviour [71]. Therefore, PI is used to test consumers' willingness to buy ceramic chopsticks of this brand.

### Continuance intention

Kim and Kang [72] believe that CI is the behaviour of users who use a particular product for a long time and regularly. It measures the likelihood that a user will continue to choose the product they are currently using in a subsequent use decision [73]. A higher CI indicates that consumers are more likely to continue using goods in the future rather than replacing them easily. At the same time, users may be influenced by unconscious reactions such as usage habits and having a natural resistance to alternative products [74]. According to the expectation confirmation model, consumers' CI on products is jointly determined by satisfaction, expectation confirmation degree and expectation after use [75]. Validation in updated and evolving technology acceptance models shows that usage habits may influence users' CI [76]. Previous studies have shown that CI is related to product characteristics. When consumers are satisfied with the ongoing use process, they may be inclined to continue to use it due to habits and conversion costs [77]. Satisfaction may also be an important factor affecting users' perception of product CI [78]. The more satisfied users are with the product, the more likely they will maintain their usage habits [79]. In addition, situational experience, visual attraction, knowledge, ability and situational experience are all important factors that may affect CI [80]. Therefore, CI is used to test whether the consumer can continue to use ceramic chopsticks at a certain frequency after purchase.

## Method

The research framework of this paper is shown in Fig 1. After extensive collection, an EP poster with high-level EPS and low-level EPS was selected to replace the original packaging. In addition, to avoid possible interference caused by brands, ceramic chopstick brands with the highest and lowest BI levels were selected. Finally, 6 research samples, including the original packaging, were made through redesign. EPS and BI were taken as classification factors, and the recovered data were analysed by two-way MANOVA. Compare the BL, PI, AND CI of consumers for these samples.

A total of 200 environmental protection-related picture samples were collected from the internet. The content of the sample is mainly to inform the public of the importance of environmental protection and advocate harmonious coexistence with nature. For example, calls for the control and reduction of emissions into the environment, including gases, refuse and sewage. Considering that respondents needed to make too much effort to classify the 200 samples directly, 30 samples were selected from all the collected picture samples through the

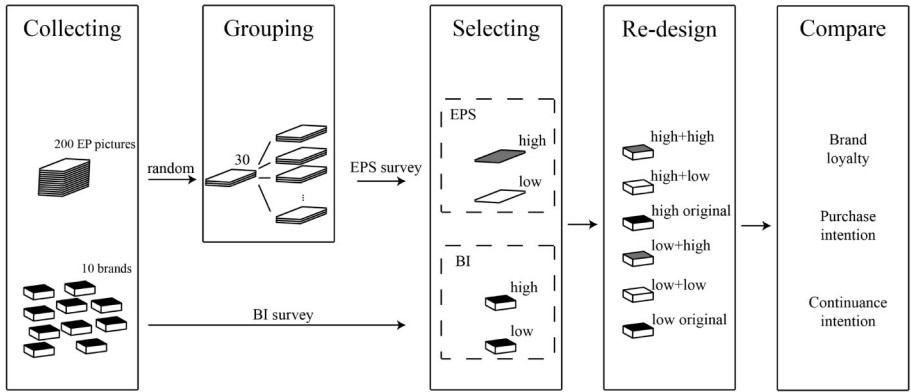

**Fig 1. Research framework.**

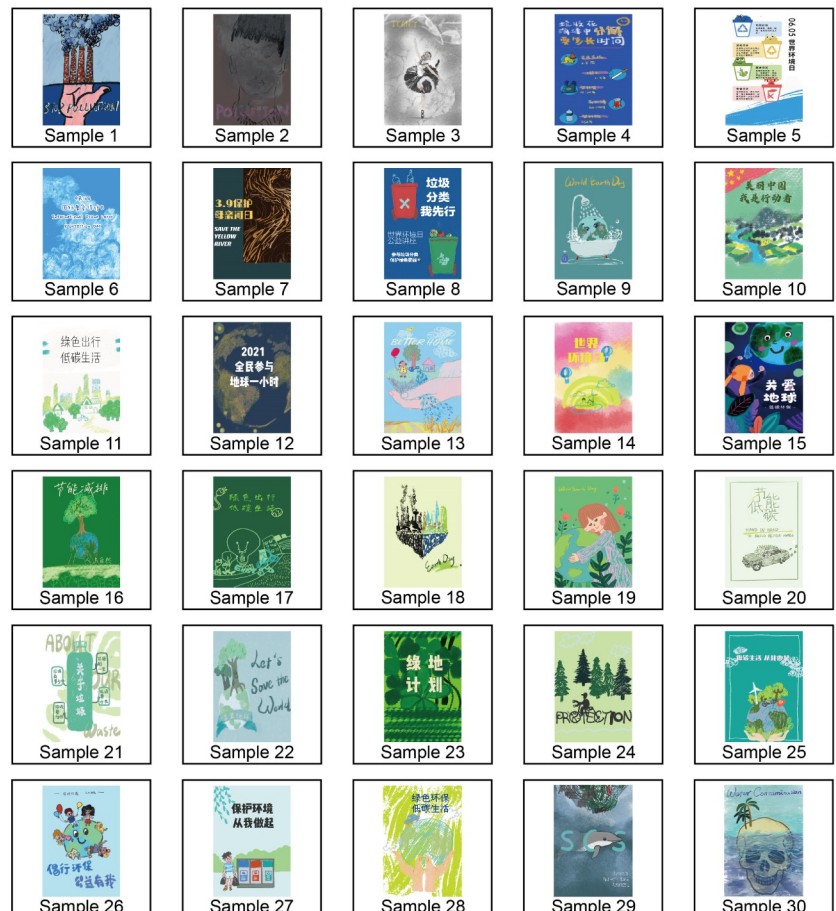

**Fig 2. Collection results of EP pictures.**

random number table to participate in the subsequent classification and design research. The 30 EP pictures randomly screened by the random number table are shown in Fig 2. The EP pictures were printed as 143.5 mm*143.5 mm single picture cards and classified through a quantitative survey. This phase of the study was completed in October 2020, and a total of 60 respondents will be investigated. There were 30 males and 30 females, mainly aged between 21 and 25. The respondents were told to make independent judgements based on their perception of sample similarity without limiting the number of groups and the number of samples placed in each group. The whole classification process does not limit the overall time, and the respondents can fully think and check the original size picture content on the computer screen at any time during the whole process. The data were statistically analysed by multidimensional scaling and cluster analysis, and 30 samples of EP pictures were divided into several groups.

After grouping, representative EP image samples were selected from each group. The selection of representative samples requires the analysis of the design advantages of the picture and the evaluation of whether the picture reflects the overall design characteristics of the group. Such judgements and generalizations require professional scrutiny, not user preference. The focus group method is a group discussion on a topic under the guidance of a trained facilitator [81]. This method is often used for in-depth understanding and solving problems [82].

**Table 1. Composition of focus group meetings.**

| Interviewee | Experience | Position | Expertise and work |
|---|---|---|---|
| *RespondentA* | 10 years | Lecture | User experience |
| *RespondentB* | 8 years | Lecture | Visual communication |
| *RespondentC* | 5 years | Department manager | Visual communication |
| *RespondentD* | 5 years | Operations director | Digital marketing |

Therefore, this paper chooses to convene experts to discuss in the form of focus groups to obtain the opinions of experts in this field. The composition of the focus group is shown in Table 1.

As few brands are selling ceramic chopsticks in China, it is difficult to collect more brands to participate in the survey. A total of 10 tableware brands selling products, including ceramic chopsticks, were collected in this stage. All brands of ceramic chopsticks can be searched and purchased through e-commerce platforms in mainland China. The collection results of ceramic chopstick brands are shown in Fig 3. It was collected in November 2020. The image used to represent the brand in the survey is taken from the actual ceramic chopstick products that the brand is selling.

Furthermore, the participants were asked to evaluate the EPS of representative poster samples and the BI of 10 ceramic chopstick brands by questionnaire survey. This phase of the study was completed in November 2020. A total of 71 participants aged between 21 and 25 were invited to participate. There were 31 males, accounting for 43.7%, and 40 females, accounting for 56.3%. The questionnaire combines two parts of case images and questionnaire items. The poster samples were presented to the subjects directly. The ceramic chopsticks brand before answering the questions, and approximately 15 minutes were spent introducing the brand of ceramic chopsticks that would appear in the study to each participant. Include the brand name, logo, brand history, country of origin, design concept and other relevant information. Information and pictures related to BI were collected through e-commerce platforms, search engines and the official website of the brand, and these contents were used as information sources. Through introductions, participants were helped to understand the brand they were not familiar with. The items used in the survey were all adapted from scales in the existing literature. The three questions proposed by Rouibah and Al-Hassan [83] were used to measure satisfaction, and the three questions proposed by Sasmita and Suki [84] were used to measure BI. As pointed out in a previous study on Malaysian consumers, the first and second items asked whether the brand image was well established, and the third item asked whether the brand had a differentiated image from other brands; these items were most vital [84]. Without changing the question method, the contents of the questionnaire were adjusted in combination with the packaging design of ceramic chopsticks. The questionnaire contains basic information on the respondents and measurements of their perceptions and preferences. A five-point isometric Likert scale (1 = strongly disagree, 5 = strongly agree) was adopted, as shown in Table 2.

The questionnaire adjustment method was used to test consumers' preferences for 6 kinds of samples formed by EPS and BI. This phase of the study was completed between December 2020 and February 2021. A total of 223 respondents, mainly aged between 21 and 25, took part in the study. Among them, 104 were males, accounting for 46.6%, and 119 were females, accounting for 53.4%. Before the survey, there were three minutes to introduce the two ceramic chopstick brands with high BI and low BI in the questionnaire to the respondents in this stage to help the respondents have a certain degree of familiarity with the brands. The test

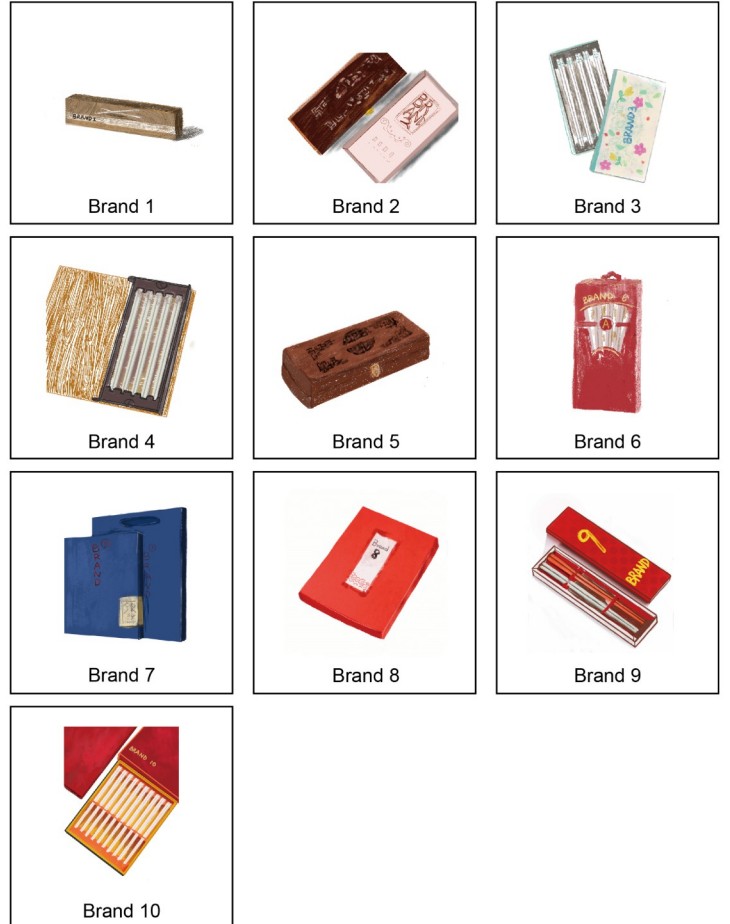

**Fig 3. Collection results of ceramic chopstick brands.**

respondents needed to evaluate the perception immediately after viewing the sample. The scale was based on a five-point Likert isometric scale (1 = strongly disagree, 5 = strongly agree). These include three questions proposed by Popp and Wilson [85] for BL and three questions proposed by Wright, et al. [86] for PI. In addition, Mäntymäki, et al. [87] proposed

**Table 2. Composition of focus group meetings.**

| Constructs | Codes | Items |
|---|---|---|
| EPS | EPS1 | 1.I was satisfied with the promoting environmental protection content I saw in this propaganda poster. |
| | EPS2 | 2.I was extremely satisfied with my overall experience of this propaganda poster in promoting environmental protection. |
| | EPS3 | 3.I was extremely pleased with my overall experience of this propaganda poster in promoting environmental protection. |
| BI | BI1 | 1.This brand of ceramic chopsticks is well established |
| | BI2 | 2.This brand of ceramic chopsticks has a clean image |
| | BI3 | 3.This brand of ceramic chopsticks has a differentiated image in comparison with the other brand. |

**Table 3. Retail ceramic chopstick consumer behaviour scale.**

| Constructs | Code | Items |
|---|---|---|
| BL | BL1 | 1.I intend to be loyal to this brand of ceramic chopsticks in the future. |
| | BL2 | 2.I will not change my affiliation from this brand of ceramic chopsticks to another brand in the future just because it is not successful anymore. |
| | BL3 | 3.I would defend this brand of ceramic chopsticks in public even if this caused problems. |
| PI | PI1 | 1.I would buy this brand of ceramic chopsticks rather than any other brand available. |
| | PI2 | 2.I am willing to recommend that others buy this brand of ceramic chopsticks. |
| | PI3 | 3.I intend to purchase this brand of ceramic chopsticks in the future. |
| CI | CI1 | 1.I intend to continue using the ceramic chopsticks I just bought for the next three months. |
| | CI2 | 2.I intend to continue using the ceramic chopsticks I just bought frequently during the next three months. |
| | CI3 | 3.I will keep on using the ceramic chopsticks I just bought in the future. |

three questions on CI. According to the paper's topic, the scale was modified into the final questionnaire by corresponding context adjustment, as shown in Table 3.

# Results

## Clustering results of EP picture

The results of graph card clustering are recycled and counted into the correlation frequency matrix, which is further converted into different frequency matrices. SPSS26 was used for data analysis. The Euclidean distance was used to establish the six-dimensional scale coordinates of the different relations between the graphs and cards through multidimensional scaling. The fitting results show that Kruskal's stress values of .096 < .10 and RSQ = .901 indicate that the model's explanatory power reaches 90.1%, and the data fitting is in an acceptable range [88]. The six-dimensional coordinates of each sample were substituted into cluster analysis. First, Ward's hierarchical cluster analysis method is adopted for calculations without specifying the number of clusters. The results show that during the countdown of the 10 agglomerates, samples involved in the early stages of agglomeration were not abnormally present. At the same time, in the process of second to last, sixth to last and ninth to last, the agglomeration coefficients increased in percentage. It increased from 14.54% to 21.27%, from 12.58% to 22.62% and from 11.90% to 14.29%, respectively. This means that a reasonable number of possible clusters might be two, six, or nine. This is shown in Fig 4.

The samples were regrouped by Ward's method of hierarchical cluster analysis in the way of specifying the number of groups. MANOVA was performed for the clustering results corresponding to the six-dimensional coordinates, and the results are shown in Table 4. When each sample was divided into two groups, there were significant differences in the results of Dimensions 1 and 2 (F>1.96, p<.05) but no significant differences in the results of Dimensions 3, 4, 5 and 6 (F>1.96, p<.05). This indicates that the clustering method does not distinguish the two groups well. However, when each sample was divided into six groups, the comparison of all dimensions achieved significant results (F>1.96, p<.05). This indicates that the group differentiation is reasonable and that there is no cross sample. Since the study samples can be clearly distinguished by six groups, there is no need to compare the results of nine groups.

Furthermore, the K-means method is used to specify the clustering results as 6 groups for cluster analysis. Cohen's kappa test was used to compare the differences in the classification results. The results showed that $\kappa$ = .75, approximate T = 8.37, and approximate significance = .00. There is no significant difference in the clustering results between the two methods, which

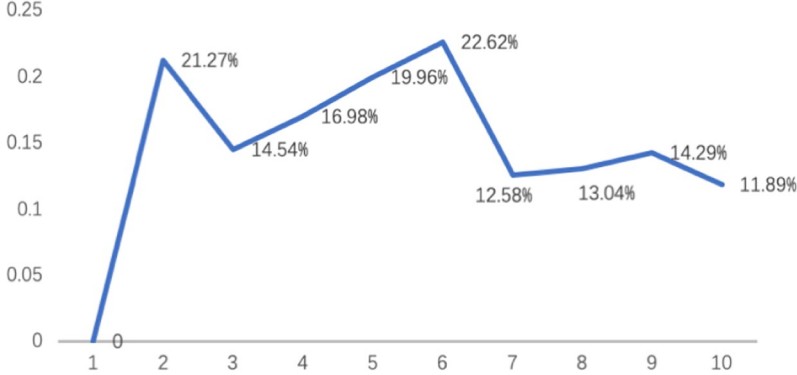

**Fig 4. Agglomeration coefficients increased percentage for the last 10 times.**

verifies the hierarchical cluster analysis results' accuracy [89]. Thus far, the result of grouping by Ward's method and specifying the number of 6 groups is adopted, as shown in Table 5.

Visualize the clustering results of picture samples into spatial images, as shown in Fig 5. The XYZ coordinate axis of space corresponds to the first three dimensions of the six-dimensional coordinates of each sample. The fourth to sixth dimensions are virtual dimensions and correspond to a change in coordinate values from small to large. The change from red to the blue of H value 0-255 in HSL is taken as dimension fourth, the change from transparent to opaque of transparency is taken as dimension fifth, and the change from small to large is taken as dimension sixth. At the same time, shape is also added as a seventh dimension to represent the results of cluster analysis, in which solid diamond represents Group 1, hollow diamond represents Group 2, hollow circle represents Group 3, square represents Group 4, solid circle represents Group 5, and cross represents Group 6. Through focus group discussions, representative samples were selected from each group, including samples 1, 5, 15, 16, 20, and 29. The main reasons for experts to select representative samples in focus group meetings are shown in Table 6.

## The questionnaire survey of EPS and BI

The test results showed that Mean = 3.70, SD = .77, Cronbach's alpha = .90 in EPS. Mean = 3.30, SD = .72, Cronbach' alpha = .77 in BI. The reliability of both constructs is greater than.60. The new reliability after the deletion of any item is lower than the current reliability, indicating that the constructs have reliability [90]. Through the observation and comparison of the representative samples of the EP Picture, the respondents' evaluation of EPS is shown in Table 7. Sample 16 had the lowest satisfaction, Mean = 3.54, SD = .73. Sample 16 and its

**Table 4. Classification rationality test.**

| | Two groups of classification | | | | | Six groups of classification | | | | |
|---|---|---|---|---|---|---|---|---|---|---|
| | Type III Sum of Squares | df | Mean Square | F | Sig. | Type III Sum of Squares | df | Mean Square | F | Sig. |
| Dim1 | 33.07 | 1 | 33.07 | 60.91 | .00 | 40.07 | 5 | 8.01 | 23.47 | .00 |
| Dim2 | 4.36 | 1 | 4.36 | 4.50 | .04 | 17.64 | 5 | 3.53 | 6.13 | .00 |
| Dim3 | .22 | 1 | .22 | .20 | .66 | 20.24 | 5 | 4.05 | 9.22 | .00 |
| Dim4 | .12 | 1 | .12 | .13 | .72 | 13.83 | 5 | 2.77 | 5.62 | .00 |
| Dim5 | .50 | 1 | .50 | .59 | .45 | 17.86 | 5 | 3.57 | 13.86 | .00 |
| Dim6 | .02 | 1 | .02 | .02 | .88 | 8.09 | 5 | 1.62 | 3.32 | .02 |

**Table 5. Clustering results of picture samples.**

| Sample | Dim1 | Dim 2 | Dim 3 | Dim 4 | Dim 5 | Dim 6 | Ward's | K Means |
|---|---|---|---|---|---|---|---|---|
| 1* | 1.848 | -1.0103 | -0.9252 | -1.2352 | -0.324 | 1.5772 | 1 | 1 |
| 2 | 2.0482 | 0.1515 | -2.442 | 0.9859 | 0.938 | 0.3297 | 1 | 1 |
| 3 | 0.6645 | -1.494 | -2.628 | -1.351 | -0.5517 | -0.6706 | 1 | 4 |
| 4 | 1.7006 | -1.3397 | 1.5757 | 1.7702 | 0.1888 | -0.5319 | 2 | 2 |
| 5* | 1.1485 | -1.9318 | 0.4025 | 2.1189 | 0.3524 | -1.022 | 2 | 2 |
| 6 | 0.4901 | 0.8495 | 0.1391 | 0.1274 | -2.3634 | 0.2716 | 3 | 3 |
| 7 | 0.5728 | 2.0819 | -0.208 | 0.149 | 0.2404 | -0.97 | 4 | 4 |
| 8 | 0.2458 | -1.4229 | 1.4382 | 0.5572 | -1.0515 | 0.2394 | 3 | 2 |
| 9 | -0.3339 | 0.8003 | -0.2628 | 1.1076 | -0.9502 | 0.9051 | 3 | 3 |
| 10 | -0.6928 | 1.5711 | 0.5396 | 0.6391 | 0.8263 | -0.1925 | 4 | 5 |
| 11 | -1.2274 | -0.5324 | 0.0539 | 0.508 | 0.8823 | 1.4199 | 5 | 5 |
| 12 | 0.5771 | 1.9054 | -1.0528 | 1.3506 | 0.2378 | 0.9751 | 4 | 1 |
| 13 | 0.7078 | 0.0344 | -0.0944 | -0.9485 | 0.7743 | -1.8896 | 4 | 4 |
| 14 | -0.2218 | 0.2427 | -0.9989 | 0.9138 | 0.0346 | -1.4276 | 4 | 4 |
| 15* | -0.028 | 0.8011 | 0.748 | 0.4953 | -1.3876 | 0.7357 | 3 | 3 |
| 16* | -0.8187 | 0.2527 | -1.1251 | -0.2802 | 0.6513 | -0.0192 | 4 | 4 |
| 17 | -1.1495 | -0.6156 | -0.5695 | -0.305 | 0.4506 | 0.7493 | 5 | 5 |
| 18 | -1.2509 | -0.4928 | 0.5771 | -0.1889 | 0.8413 | 0.5471 | 5 | 5 |
| 19 | -1.2341 | 0.7849 | -0.3535 | -0.3245 | 0.0676 | -0.4998 | 4 | 4 |
| 20* | -1.2336 | -1.3578 | -0.1558 | -0.4857 | 0.3717 | 0.2322 | 5 | 5 |
| 21 | -1.2808 | -1.2015 | 0.1438 | 0.0673 | 0.7152 | 0.1014 | 5 | 5 |
| 22 | -0.5396 | 0.1248 | -0.3317 | -0.6936 | -1.2915 | -0.397 | 3 | 3 |
| 23 | -0.8403 | 0.128 | 0.8304 | -0.9981 | 1.7011 | -0.0365 | 5 | 5 |
| 24 | -0.66 | -0.1892 | -0.0549 | -0.4793 | -1.7379 | -0.2317 | 3 | 3 |
| 25 | -0.6892 | 1.1942 | -0.1748 | -0.1897 | -0.1759 | -0.6086 | 4 | 4 |
| 26 | -1.2236 | 0.4977 | 0.8128 | -0.1642 | -0.2031 | -1.0089 | 4 | 5 |
| 27 | -1.0747 | -0.9732 | 1.0891 | 0.0086 | -0.5843 | -0.311 | 3 | 5 |
| 28 | -1.4158 | 0.3519 | 0.0972 | -0.0616 | 0.3166 | 1.1123 | 5 | 5 |
| 29* | 3.194 | 0.1337 | 1.7566 | -2.0694 | 0.1378 | 0.5051 | 6 | 6 |
| 30 | 2.7173 | 0.6554 | 1.1735 | -1.024 | 0.893 | 0.1155 | 6 | 6 |

* Representative samples of each selected group

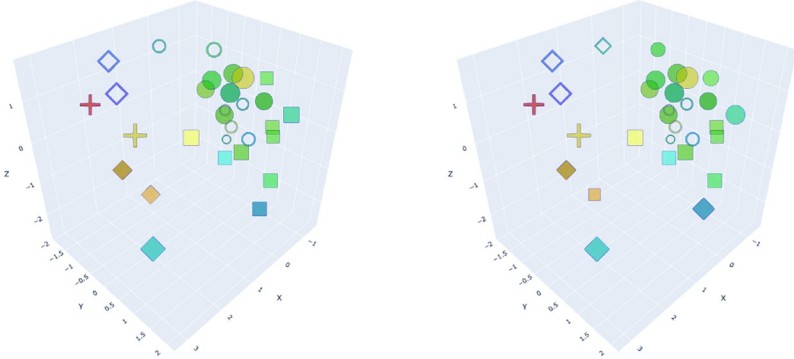

**Fig 5. Agglomeration coefficients increased percentage for the last 10 times.**

**Table 6. Experts' reasons for selecting representative samples.**

| | Expert 1 | Expert 2 | Expert 3 | Expert 4 |
|---|---|---|---|---|
| Sample 1 | Environmental elements are more intuitive | The picture combines the human hand with the polluted nature and has an impact | Most people were easy to understand about the chosen sample, other samples were too subtle or intimidating | Many pollution is Many pollution is caused by human beings, which fits the reality |
| Sample 5 | Abstention, finding it difficult to select a representative sample | Samples are more designed, and typography is better for packaging | The content of popular science covers a wider area | Additional text description, more intuitive and more convincing |
| Sample 15 | There are more environmental elements and a sense of design | Cartoonish style, not too serious | Pictures can be easily transferred to packaging, which has educational significance and good visual effect | The picture content directly appears the earth element, the environmental protection element, more intuitive |
| Sample 16 | Compared with the samples of the previous round, Chinese elements in the samples are more suitable for the subjects of this study | With the previous several rounds of style is not repeated, the colour intuitive impact | Very direct expression of environmental protection theme, clear expression of content | Show the relationship between human and nature, the main colour of green is very intuitive, excellent composition |
| Sample 20 | The style of the sample matches the overall simplicity of the group | The cars in the picture visually represent the possible environmental impact of petroleum energy | Samples link environmental protection with health, which can arouse concern about environmental protection | The publicity effect is prominent, the text proportion is large, and the green car as the main element is very intuitive |
| Sample 29 | Two of the samples were all about marine conservation themed, and the dolphins were friendlier than the other | Words and dolphins are well combined, can express human destruction and occupation of the natural environment, with humanistic concern | The image of a dolphin is more lovely and resonating | Dolphin image is more gentle expression of environmental protection, will not let the audience too panic |

representative Group 4 may be viewed as having a less acceptable design by respondents. Sample 29 had the highest degree of satisfaction, mean = 3.96, SD = .86. Sample 29 and its representative Group 6 received more positive responses and were more satisfied with the design of respondents. ANOVA was performed on samples 16 and samples 29. The homogenous test of variance showed that the Levene statistic = .781, sig = .378. Since the statistics were all less than 1.96, p<.05 indicated that the difference was not significant, which met the preassumption of data distribution and was suitable for ANOVA [91]. According to the analysis, F = 9.51,p = .00,indicating that the respondents on the two samples significantly differed in EPS. Therefore, sample 16 is suitable as a representative sample of low EPS, and sample 29 is suitable as a representative sample of high EPS.

The results showed that brand 1 of ceramic chopsticks had the lowest BI, Mean = 3.18,SD = .67. Brand 1 has a poorer BI than the other brands in the respondents' opinion. Brand 3 of ceramic chopsticks had the highest BI (mean = 3.43, SD = .75). Consumers may prefer the BI

**Table 7. Test results of EPS.**

| Groups | Representative sample | Mean | SD |
|---|---|---|---|
| 1 | sample 1 | 3.56 | .77 |
| 2 | sample 5 | 3.63 | .68 |
| 3 | sample 15 | 3.69 | .72 |
| 4 | sample 16 | 3.54* | .73 |
| 5 | sample 20 | 3.83 | .79 |
| 6 | sample 29 | 3.96* | .86 |
| Total | | 3.70 | .77 |

Note.

* The highest score sample and the lowest score sample

**Table 8. Test results of BI.**

| Brands | Mean | SD |
|---|---|---|
| 1 | 3.18* | .67 |
| 2 | 3.31 | .74 |
| 3 | 3.43* | .75 |
| 4 | 3.31 | .73 |
| 5 | 3.36 | .71 |
| 6 | 3.22 | .64 |
| 7 | 3.27 | .77 |
| 8 | 3.29 | .77 |
| 9 | 3.34 | .70 |
| 10 | 3.26 | .69 |
| Total | 3.30 | .72 |

Note.

* The highest score sample and the lowest score sample

of ceramic chopstick brand 3. The test results are shown in Table 8. ANOVA was performed on brand 1 and brand 3. The homogenous test of variance showed that the Levene statistic = .970,sig = .326,p>.05 indicates that the difference is not significant, which meets the assumption of ANOVA's data distribution. The ANOVA results showed that F = 4.35, p = .04. There were significant differences between BI, indicating that it was statistically significant to regard the two brands as high BI and low BI representatives. Therefore, ceramic chopstick brand 1 was the brand with low BI, and ceramic chopstick brand 3 was the brand with high BI.

## Research sample preparation results

A total of 3 kinds of ceramic chopstick packaging were involved in the redesign. Including the original appearance, picture sample 16 of low EPS, and picture sample 29 of high EPS. A total of two ceramic chopstick brands were involved in the redesign, including brand 1 with low BI and brand 3 with high BI. The redesign of packaging uses publicity pictures of the ceramic chopstick brand as the basis. In the modification process, the proportions of the original publicity pictures, the logo and slogan of the brand were retained. According to the colour and typesetting of the new package, the colour, size and position of the logo and slogan were adjusted to ensure observability. The result of redesign is shown in Fig 6.

## Impact of packaging on consumer behaviour

The results show that Mean = 3.61, SD = .82, Cronbach's alpha = .81 in the construct of BL. In the PI construct, Mean = 3.80, SD = .72, Cronbach's alpha = .68. In CI, Mean = 3.79, SD = .74, Cronbach' alpha = .74. Cronbach's alpha >0.6 for all constructs indicates that the items have reliability [90]. The absolute value of the skewness coefficient of each item is less than 2, and the absolute value of the kurtosis coefficient is less than 7, which is in line with the normal distribution hypothesis [92]. The correlation test results are shown in Table 9. Pearson correlation test results show that r>.60, p<.05, indicating a significant pairwise correlation between the constructs. Meanwhile, the correlation coefficient r<.90 indicates no multivariate collinearity between the constructs. Therefore, the data are subject to the assumption of a linear distribution [93].

The results of the data distribution test are shown in Table 10. The homogeneity of variance test results shows that Levene's statistic<1.96, p>.05, complying with the premise hypothesis

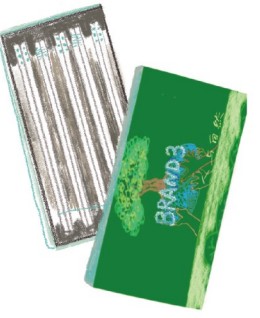
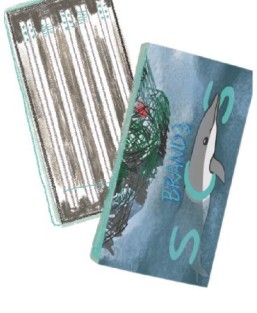
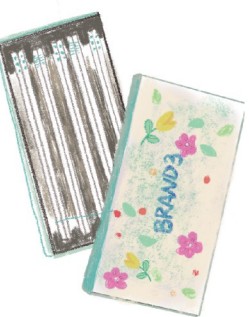

Low BI combination low EPS    Low BI combination high EPS    Original packaging of low BI

High BI combination low EPS    High BI combination high EPS    Original packaging of high BI

**Fig 6. The research samples.**

**Table 9. Results of Pearson's correlation test.**

|  | BL | PI | CI |
|---|---|---|---|
| BL | - |  |  |
| PI | .624** | - |  |
| CI | .673** | .818** | - |

Note.

$^*$p = .05;

$^{**}$p = .001.

All tests were two-tailed.

of MANOVA. The Box's Test result shows that F = 1.47, p = .05. However, Box' Test is sensitive to the number of samples, so its significance standard should be relaxed to a more relaxed level [94]. Therefore, p>.01 represents the premise assumption that covariance matrix equality is satisfied [95]. Meanwhile, the results of the interaction between EPS and BI showed that

**Table 10. Distribution test results.**

|  | Levene's Test | | | | Box' Test | | | Interaction EPS*BI | |
|---|---|---|---|---|---|---|---|---|---|
| Constructs | Statistic | df1 | df2 | Sig. | Box'M | F | Sig. | F | Sig. |
| BL | 1.49 | 5 | 1332 | .19 | 44.26 | 1.47 | .05 | 1.93 | .15 |
| PI | 1.79 | 5 | 1332 | .11 |  |  |  | 1.05 | .35 |
| CI | .99 | 5 | 1332 | .42 |  |  |  | .277 | .76 |

Note. $^*$p = .05; $^{**}$p = .001. All tests were two-tailed.

**Table 11. Pairwise test.**

| Constructs | BI | (I)EPS | (I)M (SD) | (J)EPS | (J) M (SD) | Mean Difference (I-J) | SE | Sig. |
|---|---|---|---|---|---|---|---|---|
| BL | Low | Low | 3.53 (.82) | High | 3.57 (.83) | -.04 | .08 | 1.00 |
| | | Low | 3.53 (.82) | Original | 3.58 (.77) | -.05 | .08 | 1.00 |
| | | High | 3.57 (.83) | Original | 3.58 (.77) | -.01 | .08 | 1.00 |
| | High | Low | 3.51 (.87) | High | 3.65 (.79) | -.14 | .08 | .23 |
| | | Low | 3.51 (.87) | Original | 3.78 (.80) | -.27* | .08 | .00 |
| | | High | 3.65 (.79) | Original | 3.78 (.80) | -.13 | .08 | .29 |
| PI | Low | Low | 3.79 (.74) | High | 3.86 (.74) | -.07 | .07 | .91 |
| | | Low | 3.79 (.74) | Original | 3.79 (.69) | .00 | .07 | 1.00 |
| | | High | 3.86 (.74) | Original | 3.79 (.69) | .08 | .07 | .82 |
| | High | Low | 3.75 (.80) | High | 3.91 (.67) | -.16* | .07 | .06 |
| | | Low | 3.75 (.80) | Original | 3.70 (.65) | .06 | .07 | 1.00 |
| | | High | 3.91 (.67) | Original | 3.70 (.65) | .21* | .07 | .01 |
| Ci | Low | Low | 3.74 (.78) | High | 3.88 (.75) | -.14 | .07 | .14 |
| | | Low | 3.74 (.78) | Original | 3.79 (.69) | -.05 | .07 | 1.00 |
| | | High | 3.88 (.75) | Original | 3.79 (.69) | .09 | .07 | .55 |
| | High | Low | 3.69 (.79) | High | 3.90 (.69) | -.21* | .07 | .01 |
| | | Low | 3.69 (.79) | Original | 3.76 (.69) | -.07 | .07 | .93 |
| | | High | 3.90 (.69) | Original | 3.76 (.69) | .14 | .07 | .13 |

Note. Based on estimated marginal means *. The mean difference is significant at the .05 level.

F<1.96 and p<.05 were not significant in all three constructs. This indicates that EPS and BI have a fixed effect on consumer behaviour and will not be affected by the change of the other side.

The paired comparison results are shown in Table 11. For the ceramic chopsticks brand with high BI, the BL of consumers towards the original package (M = 3.78) was significantly higher than that for the packaging design with low EPS (M = 3.51). Although not significant, it was also higher than the results of high EPS packaging (M = 3.65). This may be because, after long-term use of the original package, consumers' impression of the product package has been closely associated with brand memory. After a short contact with the new packaging design, it is difficult to integrate the brand in memory with the new packaging quickly. Consumers' PI on packaging with high EPS (M = 3.91) was significantly higher than that on packaging with low EPS (M = 3.75) and original packaging (M = 3.70). The CI of consumers on packaging with high EPS (M = 3.90) was significantly higher than that on packaging with low EPS (M = 3.69). This indicates that for ceramic chopstick brands with good BI, adding EP content with high design satisfaction into product packaging will positively effect on product sales and consumers' willingness to reuse chopsticks. Although there was no significant difference; the CI result (M = 3.90) of the high EPS package was still higher than that of the original package (M = 3.76). The reason may be that product packaging is only seen at the time of purchase and first use and gradually forgotten by consumers during the period of continued use. Therefore, compared with the original packaging with strong impression, the packaging combined with EP has a less obvious incentive effect on CI. On the other hand, for the ceramic chopstick brand with low BI, EPS also played a certain role but was relatively weak and not significant. Among the BL results of consumers for ceramic chopsticks, the original package showed the highest result (M = 3.58), slightly higher than the package with low EPS (M = 3.53) and the package with high EPS (M = 3.57). This shows that regardless of the BI level, the connection

between the long-term use of ceramic chopstick packaging and the brand may be very close in the minds of consumers. Consumers have the highest PI on packaging with high EPS (M = 3.86), and packaging with low EPS is the same as the original packaging (M = 3.79). Although the difference is not significant, EPS still shows a small gain effect on the PI in ceramic chopstick brands with low BI. In addition, consumers' CI for packaging with high EPS (M = 3.88) is slightly higher than that of the original packaging (M = 3.79), while packaging with low EPS has the lowest Intention (M = 3.74).

## Discussion

The introduction of EP into the product packaging of ceramic chopstick brands will exert a certain influence on consumers' perceptions and preferences. BI comparison results of ceramic chopstick brands show that EP packaging has similar influence trends on consumer behaviour under different BI conditions, but the magnitude of influence is different. With the improvement of BI, the difference between consumers' BL, PI and CI in different packaging of EPS expands. A detailed discussion of each construct is shown below: For BL, the inclusion of EP in packaging does not make things any better. The highest rated results were the two original packaging. The packaging combined with high EPS is slightly better than those combined with low EPS. This indicates that consumers are more loyal to the original product appearance, and the change in packaging will negatively impact the BL of consumers. Similarly, the study of Pauwels-Delassus and Descotes [96] found that a change in brand information was likely to confuse consumers, thus leading to a decline in BL. Consumers need to pay a learning cost to associate the updated information with the brands they already know. People rely on information available in their memory to make evaluations and judgements [97]. Therefore, updating product packaging may reduce the association between products and brands. The reason may be that the new packaging is temporarily changed, consumers are relatively unfamiliar with them and cannot directly correspond to the existing cognition. The accumulation of time is a necessary factor for re-establishing BL [98]. It should be noted that only the package with low EPS significantly reduced BL compared with the original package. In contrast, packaging with high EPS showed no significant negative effects. This suggests that packaging with high EPS can help effectively reduce the negative impact of changing packaging. Adding EP to the packaging of reusable tableware plays a positive role in improving PI. For brands with low BI, the PI results of packaging with low EPS are the same as those of the original packaging. In other cases, consumers all enhance their PI after EP is added to the sample package. The results explain why previous literature has shown inconsistent effects on green marketing for products as diverse as cars, mobile phones, cosmetics, cleaning products, and luxury goods [35, 37, 38, 99]. That is, the promotion effect of the green concept on marketing may be affected by BI. To obtain a greater improvement effect of purchase intent, we should pay attention to choosing packaging with a higher EPS. As Ko, et al. [100] mentioned, green marketing can help improve the product image, which is the key factor in enhancing consumers' PI. The EP in the packaging design of the ceramic chopstick brand is transformed into a positive perception of the product image in the eyes of consumers, thus enhancing the PI. There was no difference in PI between the package with lower EPS and the original package. Therefore, to improve consumer PI, reusable tableware brands can rest assured to use the method of adding EP to product packaging. The use of high EPS packaging as a design scheme can better help the marketing results achieve the expected results. The CI of consumers on ceramic chopsticks again reflects the importance of choosing high EPS packaging in product packaging. Compared with the original package, the package with high EPS brings higher CI, while the package with low EPS reduces consumers' CI. There are significant differences between the design

effects of different EPS packaging. The estimated carbon emissions show that disposable plastic tableware produces approximately 597 g/FU, and reusable plastic tableware produces only 7 g/FU [9]. Although there may be differences in the production process of plastic material, the key to determining whether it is environmentally friendly is whether the utensils can be reused enough times. Therefore, to improve consumers' CI, we should try our best to choose high EPS packaging as a design solution in the packaging design of reusable tableware. In essence, the findings offer advice on two aspects of green marketing for reusable tableware brands. The first point is the urgency for brands to improve BI. High BI brands can amplify the impact of EP packaging. However, it is difficult for brands with low BI to effectively connect the brand itself with environmental protection by using EP packaging. Therefore, consumers are not sensitive to the green marketing methods adopted by brands. This is consistent with previous research on coffee sales, which found that brand image can effectively mediate the relationship between green marketing and purchase intention [101]. Reusable tableware brands need to leverage positive BI to be effective in green marketing. Not focusing on the maintenance and improvement of BI will cause marketing work to lose meaning and value. The results of this study provide a warning to enterprises that have ignored the importance of BI for a long time. Price or quality is an important factor affecting consumer perception and decision-making [102, 103]. However, the contribution and value of BI to marketing should not be ignored [104, 105]. For the brand to effectively carry out 4P, 4C or other marketing efforts [106], it must first pay sufficient attention to improving BI and take practical actions. The second thing is that reusable tableware brands need to do careful research and judgement before deciding to use green marketing in their packaging. This is consistent with the research conclusion of Chen, et al. [107] that inappropriate green marketing may bring negative emotions of consumers to brands. Previous studies have highlighted the potential negative effects of greenwash practices. For example, greenwash negatively impacts green brand equity [108] and will reduce consumers' green purchase intention [109, 110]. However, the reusable tableware brand, which makes sustainable tableware, may not see the green marketing of its products as relevant to Green Wash. Therefore, brands relax their guard and ignore the possible negative effects of green marketing. This can lead to excessive trust and indiscriminate green marketing by brands. According to this paper's investigation results of green marketing in packaging design, packaging with lower EPS is not helpful to BL. In the case of high BI, it is significantly lower than the perceived result of the original packaging. Similarly, the package with lower EPS had no significant effect on consumer PI and CI and was slightly lower than the perceived result of the original package. The findings carry a cautionary note for reusable tableware brands. The EPS of design results should be fully considered in the green marketing of products by using the method of changing packaging. Green marketing with low EPS in product packaging is ineffective and may even bring negative effects.

## Conclusion

The addition of EP to the packaging of ceramic chopstick brand products ly positive impacts consumer behaviour. Compared with the original package, the package with high EPS can alleviate the impact of the decline in BL caused by the packaging change and can improve consumers' PI and CI to a certain extent. In contrast, packaging with low EPS has little help for consumers' PI and will also reduce BL and CI. It is helpful to choose pictures with high EPS as packaging for reusable tableware on the premise of pretesting and evaluating EPS considering both commercial effects and environmental protection. High EPS packaging can help boost retail product sales and further reduce average tableware carbon emissions by increasing consumers' intention to reuse. The research limitations have four parts. First, the respondents

who took part in the survey were selected by convenience sampling. The sample group is younger than 30 years old, mainly white-collar workers or students, and comes from the eastern part of China. No stratified sampling was conducted by different age groups or multiple regions. Second, the packaging design method in this study replaces the EP image as a whole without examining the impact of adding a separate environmental design element to the packaging on the original packaging sample. Third, the brands of ceramic chopsticks are collected from merchants that can be searched on Taobao in China, without adding the samples of brands that cannot be purchased in China. Finally, the packaging design is mainly aimed at online shopping propaganda materials, and there is no measurement of consumers' perception of real objects for offline physical packaging. Original EP pictures, pictures of ceramic chopstick brands and the research samples after redesign are replaced in this article for copyright reasons with hand-drawn figures. The figures are similar but not identical to the original image and are therefore for illustrative purposes only. Future research can further explore whether other factors interact with EPS and try to further amplify the help of package design for BL, PI and CI through the interaction of factors. In addition, the judgement basis of consumers on EPS is worth further analysis. Having the basis of judgement as a reference can help designers to make packaging with high EPS more easily.

## Supporting information

**S1 File.**
(PDF)

## Author Contributions

**Conceptualization:** Chao Gu.

**Data curation:** Jiangjie Chen, Baiwan Lv.

**Formal analysis:** Chun Yang.

**Investigation:** Jiangjie Chen, Chun Yang, Liao Jiang, Jingyue Hu, Baiwan Lv, Shuyuan Lin.

**Methodology:** Jiangjie Chen, Wei Wei, Qianling Jiang.

**Project administration:** Qianling Jiang.

**Software:** Wei Wei.

**Supervision:** Liao Jiang, Shuyuan Lin.

**Validation:** Chao Gu.

**Visualization:** Jie Sun.

**Writing – original draft:** Chao Gu, Jingyue Hu.

**Writing – review & editing:** Jie Sun, Qianling Jiang.

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
