## [Decision Letter · Decision Letter 0]

2 Nov 2021

PONE-D-21-29061The impact of Reusable Tableware packaging design combined with environmental propaganda on consumer Brand Loyalty, Purchase Intention and Continuance Intention in online retail -- Take ceramic chopsticks for examplePLOS ONE

Dear Dr. Jiang,

Thank you for submitting your manuscript to PLOS ONE. After careful consideration, we feel that it has merit but does not fully meet PLOS ONE’s publication criteria as it currently stands. Therefore, we invite you to submit a revised version of the manuscript that addresses the points raised during the review process.

We look forward to receiving your revised manuscript.

Kind regards,

Suhairul Hashim, PhD

Academic Editor

PLOS ONE

2. Please modify the title to ensure that it is meeting PLOS’ guidelines (https://journals.plos.org/plosone/s/submission-guidelines#loc-title). In particular, the title should be "specific, descriptive, concise, and comprehensible to readers outside the field" and in this case it is overly long.

5. We note that Figures 4-1, 4-4, 4-5 in your submission contain copyrighted images. All PLOS content is published under the Creative Commons Attribution License (CC BY 4.0), which means that the manuscript, images, and Supporting Information files will be freely available online, and any third party is permitted to access, download, copy, distribute, and use these materials in any way, even commercially, with proper attribution. For more information, see our copyright guidelines: http://journals.plos.org/plosone/s/licenses-and-copyright.

   1. You may seek permission from the original copyright holder of Figure(s) [#] to publish the content specifically under the CC BY 4.0 license.  

Reviewers' comments:

Reviewer's Responses to Questions

**Comments to the Author**

1. Is the manuscript technically sound, and do the data support the conclusions?

Reviewer #1: Yes

Reviewer #2: Partly

2. Has the statistical analysis been performed appropriately and rigorously? 

Reviewer #1: Yes

Reviewer #2: Yes

3. Have the authors made all data underlying the findings in their manuscript fully available?

Reviewer #1: Yes

Reviewer #2: No

4. Is the manuscript presented in an intelligible fashion and written in standard English?

Reviewer #1: No

Reviewer #2: No

5. Review Comments to the Author

Reviewer #1: Title is too long and I would suggest to amend to shorter and precise title.

There is limited explanation on underpinning theory being used in this study. Perhaps, authors include what is the underpinning theory, as Purchase Intention, Environmental Propaganda Satisfaction variables, Brand Loyalty being used.

The study also do not include proposed conceptual framework where it should come out after hypotheses development.

The literature review is too short and general.

Overall, this paper need a major revision specifically in theories, variables, hypotheses development and methodologies.

Reviewer #2: FORMATTING AND OVERALL APPEARANCE:

- The title of the manuscript is too lengthy - consider making it concise. Perhaps may remove "Take ceramic chopsticks for example".

- The manuscript is poorly written with rampant grammatical errors. Some of the sentences are not clear and do not flow well. I suggest the authors to send for professional proofreading and copy editing services.

- Avoid using is "isn't", "can't" etc. as this is a formal academic writing.

- Some in-text citations are not done according to the right format (i.e. surname of authors).

- Citations need to be updated with recent literature of within 5 years - i.e. Hamilton (1974); Stevels, Agema, & Hoedemaker, 2001; Gardner & Levy (1955) are too old.

LITERATURE REVIEW:

- Why section 3.2.2 Brand Image (BI) is suddenly being discussed together in Method section. Where are the rest of the Literature Review of the other focal constructs?

- "The marketing benefits of incorporating environmental propaganda into the Reusable Tableware packaging can be influenced by the brand." -- sentence not clear at all of what the author actually meant here.

- Need to add a section on Packaging literature review.

METHODOLOGY:

- Sampling and clustering method of ceramic chopsticks are quite confusing.

- Why focus group meetings is included rather than experimental design alone should be further justified, and results of the focus group meetings should be reported.

- Should explain the development of propaganda poster in promoting environmental protection - is this poster for experiment being guided by specific marketing model/theory on advertising? What is the process involved when you come up with the propaganda poster?

- Measurement Item for Brand Image - 2.This brand of ceramic chopsticks has a clean image -- not clear what you meant by "clean image"? This is a really confusing phrase from the marketing perspective when you are trying to describe a brand image.

- Measurement Item for Brand Image - 3. This brand of ceramic chopsticks has a differentiated image in comparison with the other brand -- not clear what is "differentiated image"? You are using an important construct from marketing, yet you have wrongly defined them. Please use proper literature on branding to help you in operationalizing brand image.

- Sampling and experiment design is being discussed in several separated sections of the manuscript (i.e. 3.1 Sample collection and clustering, 3.3 Preparation of experimental samples, 3.2.3 Experimental design, 3.4.4 Experimental design). Please consider re-arranging the experiment design description and making the paragraphs more concise.

DISCUSSION

- The overall discussion section is quite shallow and only offering a surface justification. Consider writing in a more critical manner by considering the logical connections of the constructs.

- The writing of these particular statements is very confusing with no continuity among one sentence and the next. "The accumulation of time is a necessary factor for re-establishing Brand Loyalty (Schultz & Bailey, 2000). The

strangeness of the new packaging will persist for some time, but it is inevitable when redesigning the product packaging. It should be noted that the package with low EPS significantly reduced Brand Loyalty compared with the original

package." -- not sure what you meant to express here?

6. PLOS authors have the option to publish the peer review history of their article (what does this mean?). If published, this will include your full peer review and any attached files.

Reviewer #1: No

Reviewer #2: No

---

## [Author Response · Author response to Decision Letter 0]

16 Dec 2021

Dear editor:

We have finished the revisions of the article according to the journal requirements and the suggestions of 2 reviewers. The revisions and responses to the reviewer's suggestions will be detailed in the response letter document submitted. Revisions made according to journal requirements include:

1. We adjusted the style of the article using a latex template provided by Plos One.

2. We changed the title. The new title is more specific and concise.

3 and 4. We uploaded all the data used in the article to the Dryad database with the URL: https://datadryad.org/stash/share/HQiN1DTk9HLLQE_t3JFdMmJrzvrh-DbmUNZyCEASvgI.

5. Due to the large proportion of original images were obtained from the Internet randomly, we were unable to contact all authors and obtain their permission. Therefore, we have redrawn alternative pictures with a similar style to these pictures and replaced them in the text. The statement ”Figure is similar but not identical to the original image and is therefore for illustrative purposes only“ is added at the end of the corresponding paragraph.

The affiliation information of author Chun Yang has been updated in this revision due to job changes.

In addition, as a new author (Jie Sun) has made great contributions to this article in this revision, we will add her to the author list as one of the authors of this article. This addition has been agreed by all authors of this paper, and was previously agreed by the Plos One editorial office via email.

We deeply appreciate your consideration of our manuscript, and we look forward to receiving comments from the reviewers. If you have any queries, please do not hesitate to contact me at the address below.

Response to Reviewer 1 Comments

Dear Referee,

Thank you very much for the valuable suggestions, which have made our paper more comprehensive. We have responded to each of your comments below:

Point 1: Title is too long and I would suggest to amend to shorter and precise title.

Response 1: Thank you for the suggestion. We revised the title of the article to make it more concise and specific.

Point 2: There is limited explanation on underpinning theory being used in this study. Perhaps, authors include what is the underpinning theory, as Purchase Intention, Environmental Propaganda Satisfaction variables, Brand Loyalty being used.

Response 2: Thank you for the suggestion. In Section 2, we carried out a separate literature review for each construct and supplemented the literature content of each construct. The revised part has been marked in red.

Point 3: The study also do not include proposed conceptual framework where it should come out after hypotheses development.

Response 3: Thank you for the suggestion. We added a figure in the second section to illustrate the research process of this paper (Figure 1).

Point 4: The literature review is too short and general.

Response 4: Thank you for the suggestion. In the second section, we supplemented the literature review. In this revision, more than 60 recent literatures were added and adjusted to serve as the theoretical support of this study. The revised part has been marked in red.

Response to Reviewer 2 Comments

Dear Referee,

Thank you very much for the valuable suggestions, which have made our paper more comprehensive. We have responded to each of your comments below:

Point 1: - The title of the manuscript is too lengthy - consider making it concise. Perhaps may remove "Take ceramic chopsticks for example".

Response 1: Thank you for the suggestion. We changed the title of the manuscript. The new title is more specific and concise.

Point 2: - The manuscript is poorly written with rampant grammatical errors. Some of the sentences are not clear and do not flow well. I suggest the authors to send for professional proofreading and copy editing services.

Response 2: Thank you for the suggestion. We used the Professional English Language Editing service for this revision. The certificate would be submitted in supplementary materials.

Point 3: - Avoid using is "isn't", "can't" etc. as this is a formal academic writing.

Response 3: Thank you for the suggestion. We focused on these informally written words and adjusted the presentation.

Point 4: - Some in-text citations are not done according to the right format (i.e. surname of authors).

Response 4: Thank you for the suggestion. We re-edited the references to the article with endnote.

Point 5: - Citations need to be updated with recent literature of within 5 years - i.e. Hamilton (1974); Stevels, Agema, & Hoedemaker, 2001; Gardner & Levy (1955) are too old.

Response 5: Thank you for the suggestion. We have deleted the earlier references. At the same time, the newly added literatures are all recent.

Point 6: - Why section 3.2.2 Brand Image (BI) is suddenly being discussed together in Method section. Where are the rest of the Literature Review of the other focal constructs?

Response 6: Thank you for the suggestion. We adjusted the literature review of each construct to the second section and discussed them one by one. The content of literature review is supplemented. The revised part has been marked in red.

Point 7: - "The marketing benefits of incorporating environmental propaganda into the Reusable Tableware packaging can be influenced by the brand." -- sentence not clear at all of what the author actually meant here.

Response 7: Thank you for the suggestion. It means that different brands may interfere with our evaluation of the marketing effectiveness of Reusable Tableware packaging. We have readjusted the statement to make it easier to understand. The revised part has been marked in red.

Point 8: - Need to add a section on Packaging literature review.

Response 8: Thank you for the suggestion. We have added a separate sub-section on Packaging Literature Review in section 2. The revised part has been marked in red.

Point 9: - Sampling and clustering method of ceramic chopsticks are quite confusing.

Response 9: Thank you for the suggestion. We rearranged the order of discussion in the Method section, deleted and added some content to make the research method more clear. In addition, we have added a figure in Section 3 to illustrate the research procedure (Figure 1). The revised part has been marked in red.

Point 10: - Why focus group meetings is included rather than experimental design alone should be further justified, and results of the focus group meetings should be reported.

Response 10: Thank you for the suggestion. The reason for conducting focus group interview is that when selecting representative samples of each group, we need to consider the common characteristics of the group and the design advantages of each picture, which requires professional advice. Therefore, we invited four experts to form a focus meeting for discussion. In Section 4, we supplemented the reasons for the experts' discussion and selection process of the final representative sample (Table 6). The revised part has been marked in red.

Point 11: - Should explain the development of propaganda poster in promoting environmental protection - is this poster for experiment being guided by specific marketing model/theory on advertising? What is the process involved when you come up with the propaganda poster?

Response 11: Thank you for the suggestion. Those two propaganda posters used for chopstick brand promotion are not designed by us. They were selected from existing posters through a series of classification and surveys. We adjusted the description of the selection process to make it clearer. This process includes eight steps, as follows:

1. In the initial stage of the study, 200 posters of environmental protection were collected from the Internet

2. 30 of them were selected by random numbering, and the respondents were invited to classify

3. Multivariate scale analysis was carried out on the classification results, and the 6-dimensional coordinates of each poster were calculated

4. MANOVA was used to compare the coordinates of each group to confirm the number of the six groups was optimal

5. The accuracy of clustering results was confirmed by comparing various cluster analysis methods

6. An expert focus group interview was held to select representative samples from each group

7. The environmental propaganda satisfaction of the 6 posters was investigated by questionnaire

8. There are significant differences between the two samples with the highest and lowest satisfaction which were examined by ANOVA

Point 12: - Measurement Item for Brand Image - 2.This brand of ceramic chopsticks has a clean image -- not clear what you meant by "clean image"? This is a really confusing phrase from the marketing perspective when you are trying to describe a brand image.

Response 12: Thank you for the suggestion. As for the literature on brand image, we quoted the existing scale in previous studies without changing the expression. The items are derived from Sasmita, J.; Suki, N.M. Young Consumers' Insights on Brand Equity: Effects of brand association, brand loyalty, brand awareness, and brand image. International Journal of Retail & Distribution Management 2015. In addition, because our respondents are Chinese, the clean brand image expresses the image of purity and sincerity in the Chinese context. In section 3, we supplement explanations of the original literature to illustrate the meanings of these items. The revised part has been marked in red.

Point 13: - Measurement Item for Brand Image - 3. This brand of ceramic chopsticks has a differentiated image in comparison with the other brand -- not clear what is "differentiated image"? You are using an important construct from marketing, yet you have wrongly defined them. Please use proper literature on branding to help you in operationalizing brand image.

Response 13: Thank you for the suggestion. The brand image items in the article were all quoted from the same literature, and we did not change the expression. We reexamined the original literature and differentiated image here means that consumers believe that a brand's brand image is different or special from other brands. We admit that the item we cited is just one of many literatures that attempt to define brand image. Brand image is a frequently discussed concept, and there are various descriptions about it. In Section 3, we supplement explanations of the original literature to illustrate the meanings of these items. The revised part has been marked in red.

Point 14: - Sampling and experiment design is being discussed in several separated sections of the manuscript (i.e. 3.1 Sample collection and clustering, 3.3 Preparation of experimental samples, 3.2.3 Experimental design, 3.4.4 Experimental design). Please consider re-arranging the experiment design description and making the paragraphs more concise.

Response 14: Thank you for the suggestion. In order to make the paragraphs more concise and explain the research more clearly, we have revised the organization of Section 3, including the adjustment of the sequence of paragraphs, addition and deletion of content. The revised part has been marked in red.

Point 15: - The overall discussion section is quite shallow and only offering a surface justification. Consider writing in a more critical manner by considering the logical connections of the constructs.

Response 15: Thank you for the suggestion. We have added two new paragraphs in Section 5 to try to discuss and explain in more depth. In addition, some statements are adjusted. The revised part has been marked in red.

Point 16: - The writing of these particular statements is very confusing with no continuity among one sentence and the next. "The accumulation of time is a necessary factor for re-establishing Brand Loyalty (Schultz & Bailey, 2000). The

strangeness of the new packaging will persist for some time, but it is inevitable when redesigning the product packaging. It should be noted that the package with low EPS significantly reduced Brand Loyalty compared with the original

package." -- not sure what you meant to express here?

Response 16: Thank you for the suggestion. What we want to express is that consumers need time to adapt to the new packaging, it does take a process for consumers to become familiar with the new packaging from unfamiliar and re-establish brand loyalty. We have revised this paragraph in section 5. The revised part has been marked in red.

---

## [Decision Letter · Decision Letter 1]

14 Feb 2022

The impact of reusable tableware packaging combined with environmental propaganda on consumer behaviour in online retail

PONE-D-21-29061R1

Dear Dr. Jiang,

We’re pleased to inform you that your manuscript has been judged scientifically suitable for publication and will be formally accepted for publication once it meets all outstanding technical requirements.

Kind regards,

Suhairul Hashim, PhD

Academic Editor

PLOS ONE

Additional Editor Comments (optional):

Reviewers' comments:

Reviewer's Responses to Questions

**Comments to the Author**

1. If the authors have adequately addressed your comments raised in a previous round of review and you feel that this manuscript is now acceptable for publication, you may indicate that here to bypass the “Comments to the Author” section, enter your conflict of interest statement in the “Confidential to Editor” section, and submit your "Accept" recommendation.

Reviewer #1: All comments have been addressed

Reviewer #2: All comments have been addressed

2. Is the manuscript technically sound, and do the data support the conclusions?

Reviewer #1: Yes

Reviewer #2: Yes

3. Has the statistical analysis been performed appropriately and rigorously? 

Reviewer #1: Yes

Reviewer #2: Yes

4. Have the authors made all data underlying the findings in their manuscript fully available?

Reviewer #1: Yes

Reviewer #2: Yes

5. Is the manuscript presented in an intelligible fashion and written in standard English?

Reviewer #1: Yes

Reviewer #2: Yes

6. Review Comments to the Author

Reviewer #1: (No Response)

Reviewer #2: - The authors have made great efforts in revising the manuscript. Proper English usage is used throughout the manuscript signifying that a professional proofreading service has been engaged.

- The paragraphs especially for literature review, methods and findings are now in good shape that they are being laid out in a more structured manner that may encourage ease of reading.

- A lot of confused statements/claims in the previous version have been removed and flow of ideas seem improved.

- References have been updated with the recent ones.

- Overall, I am quite satisfied with all the revisions made and would recommend this manuscript to be accepted for publication.

7. PLOS authors have the option to publish the peer review history of their article (what does this mean?). If published, this will include your full peer review and any attached files.

Reviewer #1: No

Reviewer #2: **Yes: **Zuraidah Sulaiman

---

## [Editor Report · Acceptance letter]

2 Mar 2022

PONE-D-21-29061R1 

The impact of reusable tableware packaging combined with environmental propaganda on consumer behaviour in online retail 

Dear Dr. Jiang:

I'm pleased to inform you that your manuscript has been deemed suitable for publication in PLOS ONE. Congratulations! Your manuscript is now with our production department. 

Kind regards, 

on behalf of

Dr. Suhairul Hashim 

Academic Editor

PLOS ONE